# Do LLMs Align with My Task? Evaluating Text-to-SQL via Dataset Alignment

## Abstract

Supervised Fine-Tuning (SFT) is an effective method for adapting Large Language Models (LLMs) on down-stream tasks. However, variability in training data can hinder a model's ability to generalize across domains. This paper studies the problem of *dataset alignment* for Natural Language to SQL (NL2SQL or text-to-SQL), examining how well SFT training data matches the structural characteristics of target queries and how this alignment impacts model performance. We propose the Alignment Ratio (AR), a metric that quantifies structural alignment and can serve as a predictive, actionable decision criterion to select or filter fine-tuning datasets. We hypothesize that alignment can be accurately estimated by comparing the distributions of structural SQL features across the training set, target data, and the model's predictions prior to SFT. Through comprehensive experiments on three large cross-domain NL2SQL benchmarks and multiple model families, we show that structural alignment is a strong predictor of fine-tuning success. When alignment is high, SFT yields substantial gains in accuracy and SQL generation quality; when alignment is low, improvements are marginal or absent. These findings highlight the importance of alignment-aware data selection for effective fine-tuning and generalization in NL2SQL tasks.

## 1 Introduction

Natural Language to SQL—the automatic conversion of user queries into executable SQL commands—enables non-technical users to interact with databases using natural language, simplifying access to relational databases without requiring the knowledge of SQL syntax or schema details. NL2SQL is expected to be an important tool in many industries, from business intelligence to healthcare and education. Traditional NL2SQL models relied heavily on syntactic and semantic parsing, but recent advancements in transformer-based models have drastically improved the accuracy and robustness of these systems (Gao et al., 2023; Pourreza & Rafiei, 2024a).

While NL2SQL models have achieved impressive results on benchmarks, they often struggle in real-world settings due to the variability of natural language inputs and diversity in query structures and database schemas. To be effective across domains, models must generalize beyond their training data—a task made difficult by the complexity of both natural language and SQL. Transfer learning, especially through supervised fine-tuning (SFT), has emerged as a promising solution, enabling models to adapt to new tasks or domains by leveraging labeled data from related sources (Zoph et al., 2016; Min et al., 2017; Sun et al., 2024). In NL2SQL, SFT allows models to learn domain-specific patterns, improving performance even when source and target datasets differ significantly. However, challenges remain: fine-tuned models may overfit or fail to transfer knowledge effectively when alignment between datasets is poor.

As an example, consider fine-tuning CodeLlama-7B (Roziere et al., 2023) for the task of NL2SQL, aiming to improve its performance on the Gretel development set (§ 4.1). As shown in Figure 1, the model's execution accuracy after fine-tuning can improve, remain unchanged, or even deteriorate. A key question is if the post-SFT performance of a model on a target dataset can be predicted beforehand. Such predictions would be invaluable in identifying datasets that could potentially improve performance or deciding if it is not worth investing time and resources in fine-tuning when no suit-

able data is available. Several factors influence a model's post-SFT performance on a target dataset, including the patterns it was exposed to during pretraining, the relevance of the fine-tuning data to the target dataset, and the model's overall generalizability. Given the limited public information about large language models, predicting performance remains a complex challenge.

Two closely related problems explored in the literature are data selection for SFT (Xie et al., 2024; Kang et al., 2024; Albalak et al., 2024), aimed at reducing the size of training data to improve efficiency and scalability, and evaluating the impact of SFT across various models and tasks (Ding et al., 2023; Sun et al., 2024; Pourreza & Rafiei, 2024b). Both lines of work assume that relevant training data is available and provided. In this paper, we relax this assumption to explore how well a source training dataset aligns with the LLM's background knowledge from pretraining and the target datasets on which the model will be evaluated. This relaxation is particularly important when the source and target datasets largely differ, or when multiple datasets are available for selection. Our extensive experiments on three large cross-domain NL2SQL datasets using different model sizes from three LLM families, QWen2, CodeLlama and Qwen2.5-coder-instruct, show that this alignment can be detected in most cases, and our approach accurately quantifies it across different models within the same family.

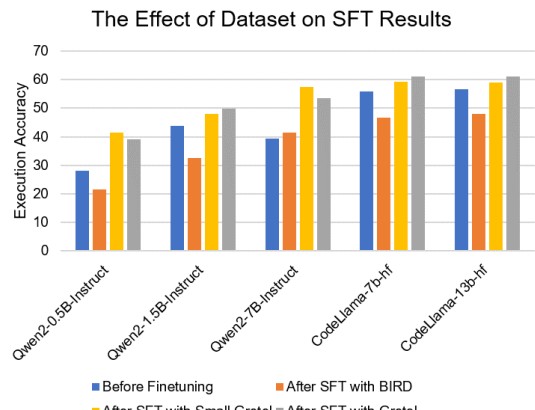

Figure 1: Execution accuracy of various models on the Gretel test set before and after supervised fine-tuning (SFT) with different datasets. The graph highlights performance variability, showing instances of accuracy improvement and degradation across the datasets and potential benefits of post-SFT performance prediction.

This paper makes key contributions to the study of dataset alignment in NL2SQL tasks by examining how structural relationships between training and target datasets affect model performance during supervised fine-tuning. Through extensive evaluation on NL2SQL benchmarks, we show that well-aligned SFT data significantly enhances model accuracy and query generation, while misaligned data impairs performance, highlighting the importance of alignment in transfer learning. Furthermore, we propose and validate a predictive framework for assessing the alignment between SFT training and target datasets, enabling informed dataset selection and reducing the risks of fine-tuning on poorly aligned data. Our framework demonstrates that Alignment Ratio (AR) is a predictive, actionable metric, which can prevent harmful fine-tuning and prioritize datasets that improve performance.

Our contributions can be summarized as follows:

- **Dataset Alignment in NL2SQL**: We introduce and systematically investigate the role of structural alignment between training and target datasets in supervised fine-tuning for NL2SQL tasks.

- **Empirical Evaluation Across Benchmarks**: Through comprehensive experiments on NL2SQL benchmarks using LLMs from QWen and CodeLlama families, we demonstrate the strong correlation between alignment quality and model performance.

- **Predictive Framework for Alignment Assessment**: We develop and validate an approach to predict the post-SFT performance based on dataset alignment, enabling proactive selection of training data and mitigating the risk of performance degradation. This approach provides a concrete decision-making tool for practitioners, extending prior KL-based methods with actionable insights.

## 2 RELATED WORK

**Data Selection for Continued Pretraining or Fine-Tuning** Importance Resampling (Xie et al., 2024) introduces *KL reduction*, a KL-divergence-based metric measuring how selected data shifts a model toward the target distribution in a hashed n-gram feature space. This work shows strong correlation between reduced divergence and downstream performance, demonstrating that distributional similarity can guide data curation. Optimal Transport Distance (Kang et al., 2024) similarly aligns pretraining distributions to a target domain but assumes access to the full pretraining corpus. Broader surveys highlight diversity, quality, and domain relevance as key criteria in data selection (Albalak et al., 2024). However, most such methods have not been evaluated on structured compositional tasks like NL2SQL, where syntactic patterns and schema grounding play a central role.

Recent 2025 works have explored adaptive SFT, curriculum learning for reasoning models, and representation-level alignment to select exemplars (e.g., (Zhu et al., 2025; Caferoğlu et al., 2025; Zhang et al., 2025)). While these methods optimize fine-tuning procedures, they primarily do not predict whether SFT will improve performance before training.

Prior KL-based approaches such as (Everaert & Potts, 2024; Kurian & Allali, 2024) use divergence metrics for subset selection or drift monitoring, but they do not establish a predictive relationship between dataset alignment and post-SFT gains. Our structural KL-alignment approach addresses this gap.

**Data Selection for Fine-Tuning Code Generation Models** Data selection remains underexplored in code generation despite substantial progress in fine-tuning LLMs. Tsai et al. (2024) propose *Code Less, Align More*, which prunes redundant training examples to maintain task alignment while reducing dataset size. Liu et al. (2024) introduce multitask fine-tuning to improve generalization, and Li et al. (2022) leverage large curated datasets for competition-level code generation. Parameter-efficient SFT approaches such as Samo et al. (2024) demonstrate improved efficiency for Python code generation.

Additionally, emerging Text-to-SQL works such as SPFT-SQL (Zhang et al., 2025), SING-SQL (Caferoğlu et al., 2025), PARSQL (Dai et al., 2025), and Solid-SQL (Liu et al., 2025) further highlight the importance of dataset selection, synthetic data generation, and structural alignment in fine-tuning pipelines. These studies underscore that careful dataset design significantly impacts performance and efficiency, yet they do not quantify alignment metrics predictive of downstream gains.

**The Role of Supervised Fine-Tuning in NL2SQL Performance** SFT is widely used to enhance NL2SQL accuracy (Sun et al., 2024; Scholak et al., 2021; Pourreza & Rafiei, 2024b), and recent advances in data synthesis further boost performance. Yang et al. (2024b) combine weak and strong LLMs to generate diverse training data, while Pourreza et al. (2024b) use synthetic data to address dialect gaps through model merging. Factors influencing transfer success include SQL structural complexity (Pourreza et al., 2024a), linguistic diversity of natural language inputs (Ning et al., 2022), and schema variability (Li et al., 2024).

Curriculum-based approaches such as (Zhu et al., 2025) structure training data by SQL complexity, highlighting the benefits of controlling dataset structure for SFT performance. Nonetheless, none of these studies directly use predictive alignment metrics to guide dataset selection or forecast SFT success, which is the focus of our work.

## 3 METHODOLOGY

### 3.1 SFT FOR TEXT-TO-SQL

SFT for text-to-SQL entails training a large language model on a dataset $T = (q, s, D)$, where each triplet comprises a natural language question $q$, its corresponding SQL query $s$, and the associated database schema $D$. Since it is typically unknown which tables are pertinent to a specific query, $D$ encompasses all database tables, enabling the model to learn to identify the relevant tables. The goal

is to minimize the empirical loss:

$$\min_{\phi} \frac{-1}{|T|} \sum_{(q,s,D) \in T} \sum_{t=1}^{|s|} log[Pr_{\phi}(s_t|D, q, s_{1,...,t-1})], \qquad (1)$$

where $Pr_{\phi}$ denotes the probability of generating the next SQL token $s_t$ given the database $D$, question $q$, and the previously generated token sequence $s_{1,...,t-1}$. As the model weights $\phi$ are updated, the predictions are expected to align more closely with the training data. To ensure that these improvements generalize to unseen data, it is crucial that the training data $T$ is representative of the target data $G$. The change in prediction for an input $x$, as the model moves from initial weights $\phi_0$ to updated weights $\phi$, can be quantified in terms of the difference between the two probability distributions $Pr_{\phi}(.|x)$ and $Pr_{\phi_0}(.|x)$.

Let $L_T$ and $L_G$ represent the language models of the training and target test data, respectively, defined as probability distributions over word sequences, while $M$ and $M'$ denote our LLM before and after fine-tuning on $T$. By design, $M'$ cannot be farther from $L_T$ than $M$, as this would imply that the loss has not been minimized. However, we want to assess whether fine-tuning will bring the base model M closer to the language model of the target $L_G$. Direct comparison between the language model of M and $L_G$ is challenging because $M$ operates over the entire vocabulary and larger contexts, whereas $L_G$ is limited to a smaller set of tokens and contexts specific to the target data $G$. To bridge this gap, we generate outputs from $M$ on dataset $G$, resulting in a set of SQL queries. Let $L_{M,G}$ represent the language model of those generated queries. If $L_T$ is *farther* from $L_G$ than $L_{M,G}$, fine-tuning $M$ on $T$ may inadvertently move the model away from $G$, potentially diminishing performance on the target data. Next, we examine query templates as critical features for comparing $L_T$, $L_G$ and $L_{M,G}$.

## 3.2 DERIVING STRUCTURAL QUERY TEMPLATES

The process of generating SQL queries from natural language questions typically involves parsing the question, identifying relevant tables and columns, selecting an appropriate SQL query template, and filling in the details. While foundational steps, such as question parsing, are covered during the training of LLMs, the limited size and diversity of task-specific datasets (in our case, text-to-SQL) reduce the likelihood of exposing models to the wide range of structural query patterns required for effective inference. We hypothesize that SFT data bridges this gap by introducing critical structural variations. This aligns with observations in selecting in-context examples, where examples with similar query structures to the target yield the greatest benefit Pourreza et al. (2024a). Building on this, we focus on structural features in the form of query templates learned from SFT data.

To derive these templates, SQL queries are parsed and schema-specific token sequences—commonly found at the leaves of the parse tree—are removed. These include table and column names that vary across databases, as well as literals that do not affect the query logic, as shown in Appendix §A.

## 3.3 MEASURING DATASET ALIGNMENT AND PREDICTING POST-SFT PERFORMANCE

A metric to assess the alignment between an SFT dataset, $D_{SFT}$, and a target dataset, $D_{target}$, is the proportion of distinct query templates in $D_{target}$ that also appears in $D_{SFT}$, which we refer to as OVLP ratio. However, since long query templates often share similar structures without being identical, we adopt a more granular metric based on n-gram features of query templates. Specifically, n-grams of lengths ranging from 1 to $l_{max}$ tokens are extracted from each dataset, where $l_{max}$ is bounded by the length of queries. To ensure the quality and relevance of these n-grams, we exclude those that lack SQL keywords, begin or end with commas, or have unmatched parentheses. The remaining n-grams and their frequencies are then used to represent the distribution of each query set for further analysis.

To quantify the differences between n-gram distributions of $D_{SFT}$ and $D_{target}$, we utilize KL-divergence, a metric widely used in reinforcement learning from human feedback (RLHF) to maintain policy proximity Bai et al. (2022); Sessa et al. (2024) and in knowledge distillation to align token distributions between student and teacher models Wu et al. (2024). The KL divergence is

defined as:

$$D_{\mathrm{KL}}(P \parallel Q) = \sum_i P(i) \log \frac{P(i)}{Q(i)}, \tag{2}$$

where $P$ and $Q$ represent the n-gram probability distributions of $D_{\mathrm{SFT}}$ and $D_{\mathrm{target}}$, respectively. This metric quantifies the divergence between datasets, helping track shifts in token generation after fine-tuning and indicating whether the output distribution aligns with the target.

To convert divergence into a measure of alignment, we define the KL-alignment metric: $A_{\mathrm{KL}}(P,Q) = exp(-D_{KL}(P||Q)/c)$. This metric ranges from 0 to 1, where 1 indicates perfect alignment (achieved when $D_{\mathrm{KL}}(P \parallel Q) = 0$). The constant $c$ serves as a scaling factor to bound the alignment scores from below.

We hypothesize that for SFT to achieve potential performance improvements, the training dataset must align more closely with the target dataset than the baseline model's distribution. Misalignment between the training data and the target can limit improvements or even degrade performance.

To quantify how well SFT data aligns with the target relative to the baseline model in feature space, we introduce the *alignment ratio* (AR), defined as the ratio between $\mathrm{A}_{KL}(\bar{D}_{\mathrm{target}} \parallel \bar{D}_{\mathrm{train}})$ and $\mathrm{A}_{KL}(\bar{D}_{\mathrm{target}} \parallel \bar{D}_{\mathrm{pred}})$, where $\bar{D}_{\mathrm{train}}$ and $\bar{D}_{\mathrm{target}}$ represent the empirical feature distributions of the training and target datasets, respectively, while $\bar{D}_{\mathrm{pred}}$ denotes the feature distribution of the baseline model's predictions on the target dataset. A higher alignment ratio ($AR_{KL} > 1$) indicates that the training dataset aligns better with the target than the baseline model, signaling potential for post-SFT performance improvement.

## 4 EXPERIMENTAL EVALUATION

### 4.1 DATASETS

We evaluate our proposed approach on three NL2SQL datasets with different complexities. **BIRD** (Li et al., 2024) includes 95 real-world databases from 37 domains, featuring complex queries involving multiple table joins, nested subqueries, and operations requiring both deep schema understanding and natural language comprehension. It includes 9,428 training samples and 1,534 development samples. **Spider** Yu et al. (2018) contains 10,181 questions over 200 databases, with 140 used for training and the remaining 60 reserved for development and testing. **Gretel** (Meyer et al., 2024) is a large-scale synthetic dataset that spans 100 distinct domains and includes 100,000 training samples and 5,850 test samples. **SmGretel** is a size-controlled subset of Gretel, randomly sampled to match the training set size of BIRD, allowing for controlled comparisons across datasets without confounding effects due to dataset scale.

### 4.2 MODELS

In our experiments, we evaluate the performance of various models from the Qwen (Yang et al., 2024a), Llama-2 (Touvron et al., 2023), and Deepseek (Guo et al., 2024) families, encompassing a range of sizes and capabilities. Specifically, we investigate the following model configurations: Qwen2 0.5B, Qwen2 1.5B, Qwen2 7B, Codellama 7B, Codellama 13B, Deepseek-coder 6.7B, Qwen2.5-coder-instruct 3B, Qwen2.5-coder-instruct 7B, and Qwen2.5-coder-instruct 14B. Each model undergoes various training strategies, including supervised fine-tuning (SFT) and few-shot learning. Unless stated otherwise, our analysis employs zero-shot prompting. Details of prompts used for SFT and few-shot tasks are detailed in Appendix §E.

### 4.3 EVALUATION METRICS AND EXPERIMENTAL SETTINGS

We evaluate model performance using two standard metrics: execution accuracy (EX) and exact match (EM), commonly adopted in benchmarks like BIRD (Li et al., 2024) and Spider (Yu et al., 2018). *Execution accuracy* measures whether the predicted and ground-truth SQL queries yield identical results when executed on a database, regardless of syntactic differences. *Exact match* compares each clause of the predicted query to the corresponding clause in the ground truth, treating them as sets. A prediction is correct only if all components match exactly. In addition, we assess dataset alignment using the KL-alignment metric described in §3.3.

For reproducibility, we set the maximum n-gram length $l_{max} = 15$, based on the average query length across datasets (14.30 tokens on BIRD dev, 12.70 on Gretel test), which also aligns with model-generated outputs. The constant $c$ in KL-alignment was chosen to lower-bound the score at $1/e$. Additional details, including the OVLP ratio metric, are provided in Appendix §B.

Table 1: KL-alignment scores of base models and training datasets with the Spider dev, BIRD Dev, and Gretel Test target sets. Higher scores indicate greater syntactic structural alignment with target sets.

| Model/Dataset | Spider | BIRD | Gretel |
|---|---|---|---|
| CodeLlama 13B | 0.52 | 0.51 | 0.64 |
| CodeLlama 7B | 0.58 | 0.49 | 0.68 |
| QWen2 7B | 0.63 | 0.61 | 0.68 |
| QWen2 1.5B | 0.61 | 0.60 | 0.69 |
| QWen2 0.5B | 0.46 | 0.57 | 0.66 |
| Deeepseek 6.7B | 0.59 | 0.53 | 0.67 |
| Qwen2.5-coder 14B | 0.80 | 0.67 | 0.71 |
| Qwen2.5-coder 7B | 0.61 | 0.66 | 0.72 |
| Qwen2.5-coder 3B | 0.76 | 0.67 | 0.71 |
| Spider Train | 0.81 | 0.46 | 0.43 |
| BIRD Train | 0.49 | 0.74 | 0.42 |
| Gretel Train | 0.61 | 0.52 | 0.88 |
| SmGretel Train | 0.46 | 0.44 | 0.71 |

## 4.4 ALIGNMENT ACROSS DATASETS AND MODELS

Table 1 presents KL-alignment scores for our models on the the development sets of Spider and BIRD, and the test set of Gretel. With few exceptions, alignment scores are highest on Gretel, followed by Spider, and lowest on BIRD. This trend reflects the relative difficulty of the benchmarks for the tested models—BIRD poses the greatest challenge, while Gretel is the easiest. The newer Qwen2.5-coder models perform strongly across all three datasets, consistent with findings from prior work (Hui et al., 2024).

These results confirm that KL-alignment is an effective measure of syntactic similarities across datasets and models, underscoring the importance of training on data that closely matches the target syntax to optimize model performance. Further analysis of query template overlap (Appendix §C) provides additional support for these observations.

## 4.5 CHANGE IN ALIGNMENT AFTER SFT

Table 2 demonstrates several noteworthy trends, in terms of change in KL-alignment after SFT.

**Post-SFT Improvements on BIRD**  All models show a clear increase in KL-alignment on BIRD after fine-tuning on its training data. The largest gains are observed in the CodeLlama models (+0.14 to +0.15), indicating that these models benefit substantially from task-specific supervision to improve syntactic alignment with BIRD queries. Similar results are obtained on Gretel (See Appendix §D for more details).

**Trade-offs on Other Datasets**  Several models exhibit reduced alignment with Gretel and Spider after BIRD fine-tuning, particularly CodeLlama 7B (–0.22 on Gretel) and 13B (–0.17 on Spider). This suggests that fine-tuning on a single dataset can result in overfitting to its structure, reducing generalization to other schema distributions.

**Qwen2.5 Models Are More Stable**  The newer Qwen2.5 models exhibit high base KL-alignment scores across all datasets and show minimal change post-fine-tuning (mostly between –0.01 and +0.05). This stability suggests that these models are inherently well-aligned with the syntactic patterns in all three datasets, and less sensitive to further fine-tuning.

Table 2: KL-alignment scores of model outputs (zero-shot) across three datasets—BIRD (dev), Gretel (test), and Spider (dev)—before and after SFT on the BIRD training set. Left: base KL-alignment values. Right: changes in alignment scores ($\Delta$) post-SFT, indicating how fine-tuning on BIRD affects alignment with each dataset.

| Model | Base KL-Alignment | | | $\Delta$ After SFT on BIRD | | |
|---|---|---|---|---|---|---|
| | **BIRD** | **Gretel** | **Spider** | **BIRD** | **Gretel** | **Spider** |
| CodeLlama 13B | 0.51 | 0.64 | 0.52 | +0.15 | -0.11 | -0.17 |
| CodeLlama 7B | 0.49 | 0.68 | 0.58 | +0.14 | -0.22 | +0.23 |
| Qwen2 7B | 0.61 | 0.68 | 0.63 | +0.05 | -0.11 | +0.00 |
| Qwen2 1.5B | 0.60 | 0.69 | 0.61 | +0.04 | -0.12 | -0.10 |
| Qwen2 0.5B | 0.57 | 0.66 | 0.46 | +0.06 | -0.08 | +0.01 |
| Qwen2.5-coder-instruct 14B | 0.67 | 0.71 | 0.80 | -0.01 | +0.00 | +0.05 |
| Qwen2.5-coder-instruct 7B | 0.66 | 0.72 | 0.61 | +0.01 | +0.00 | +0.04 |
| Qwen2.5-coder-instruct 3B | 0.67 | 0.72 | 0.76 | +0.00 | +0.00 | -0.02 |

## 4.6 Effect of Few-shot Prompting on Alignment

Few-shot prompting is a widely used technique to influence model outputs without extensive fine-tuning. To evaluate its impact on alignment, we tested three configurations: zero-shot prompting (None), few-shot prompting with ExS1, and few-shot prompting with ExS2. Each few-shot setting included three in-context examples from the training datasets. ExS1 contained one query template shared with the target datasets, while ExS2 included two shared templates. Details about these examples are in Appendix §E.

As shown in Table 3, KL-alignment scores remain largely stable across all prompting settings, for both base and fine-tuned (SFT) models. Base models show a marginal increase in alignment scores from 0.61 (zero-shot) to 0.63 (ExS2), suggesting that the inclusion of more relevant examples may offer minor syntactic guidance. However, the improvements are small and within standard deviation ranges, indicating no substantial shift in alignment behavior. SFT models show near-identical scores across all configurations, implying that prior supervised training likely overrides any influence from in-context examples.

Table 3: KL-alignment scores (mean ± standard deviation) for base and fine-tuned (SFT) models under zero-shot and few-shot prompting. Few-shot settings include ExS1 (one shared query template with target datasets) and ExS2 (two shared query templates). Results highlight the limited impact of few-shot prompting on alignment across models and datasets.

| Few-shot Setting | Base Models (Mean ± SD) | SFT Models (Mean ± SD) |
|---|---|---|
| Zero-shot | 0.61 ± 0.07 | 0.61 ± 0.10 |
| Few-shot (ExS1) | 0.62 ± 0.06 | 0.61 ± 0.08 |
| Few-shot (ExS2) | 0.63 ± 0.04 | 0.61 ± 0.08 |

## 4.7 KL-Alignment vs. Model Accuracy

Figure 2 illustrates the relationship between KL-alignment and two key evaluation metrics—execution accuracy (top) and exact match accuracy (bottom)—for base models across zero-shot and few-shot prompting settings.

Across both panels, a consistent positive correlation emerges: models with higher KL-alignment scores tend to achieve better performance on both execution and exact match metrics. This trend holds across model families and datasets, underscoring KL-alignment as a useful proxy for measuring syntactic compatibility and downstream SQL generation quality.

Notably, Qwen 2.5 Coder models (gray) demonstrate strong alignment and accuracy, dominating the upper-right regions in both plots. In contrast, CodeLlama models (orange) show lower KL-

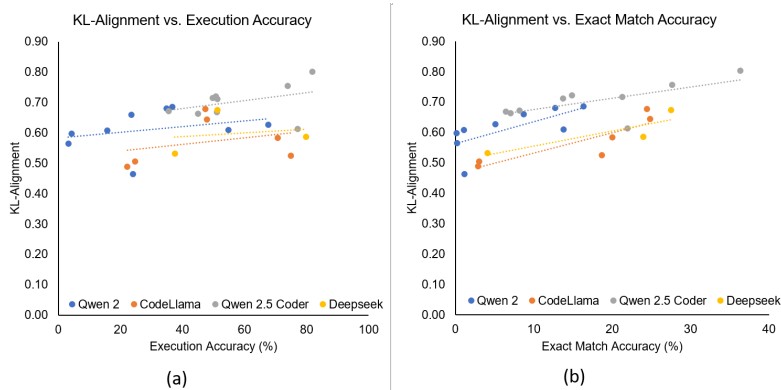

Figure 2: Correlation between KL-Alignment and a) Execution Accuracy and b) Exact Match Accuracy for base model outputs. Higher KL-Alignment generally corresponds to improved execution accuracy across model families.

alignment and accuracy, indicating less syntactic consistency with target datasets. Qwen 2 (blue) and Deepseek (yellow) occupy intermediate positions, with Qwen 2 models showing slightly better alignment on average.

## 4.8 PREDICTIVE CAPABILITY OF ALIGNMENT RATIO

While increasing KL-alignment typically benefits model performance, it is equally important to identify when fine-tuning may degrade a baseline model. To this end, we investigate whether the Alignment Ratio (AR)—introduced in §3.3—can serve as a predictor of post-SFT model accuracy.

Figure 3 plots AR values against the percent change in execution accuracy after SFT. A clear trend emerges: datasets with AR>1 generally lead to accuracy improvements, whereas datasets with AR<1 often result in limited or negative performance change, indicating that AR can guide dataset selection before fine-tuning.

This predictive relationship is strongest in CodeLlama models (r=0.624, p=0.030), and also statistically significant for Qwen-2 models (r=0.540, p=0.037), though somewhat weaker (r=0.540, p=0.037), though somewhat weaker. This difference may be due to the smaller size of the Qwen models (0.5B–7B) compared to CodeLlama (7B and 13B), which may make them more adaptable to moderately misaligned data.

In practice, AR can be operationalized as follows: if a candidate SFT dataset has AR > 1, it is likely to yield accuracy improvements and can be selected as-is; if AR < 1, the dataset should be filtered, down-weighted, or excluded. Applying SFT only when AR > 1 would have improved accuracy across all evaluated models except Qwen2.5-Coder, which shows minimal headroom for SFT.

In contrast, Qwen2.5-Coder models exhibit no meaningful correlation (r=0.029, p=0.941). These models are already highly capable on NL2SQL tasks due to strong pretraining, leaving minimal room for improvement through SFT. Their post-SFT accuracy varies by less

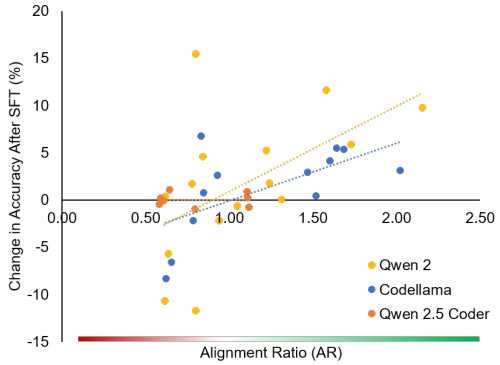

Figure 3: Predictive nature of alignment ratio (AR): Datasets with AR > 1 generally show accuracy improvement after SFT, while those with AR < 1 exhibit similar or decreased accuracy. The colour bar at the bottom of the figure highlights better (dark green) and poorer (dark red) alignment ratios.

than ±1%, with AR values clustered between 0.5 and 1.1, suggesting limited utility of AR as a predictor for such models.

Overall, these results highlight that while AR is a strong predictor of SFT success, it should be treated as a guidance signal rather than a strict guarantee.

### 4.9 ESTIMATING TARGET QUERY DISTRIBUTION FROM SMALL SAMPLES

In industry settings, user query logs are commonly used for training and evaluation. Our method provides a cost-efficient way to reorganize such logs by estimating target query structure alignment from small samples—without requiring new annotations.

As shown in Table 4, KL-alignment estimated from a small query sample closely mirrors the trends seen in full datasets (see Table 2 for SFT on BIRD and Table 7 for SFT on Gretel). Fine-tuning on Gretel increases alignment with its test distribution across all models while often decreasing alignment with BIRD, especially for smaller QWen models. Conversely, SFT on BIRD improves BIRD alignment but harms Gretel alignment, underscoring the asymmetry of cross-domain generalization. These results show that small samples are sufficient to guide fine-tuning decisions and predict domain-specific alignment effects.

Table 4: KL-alignment of model-generated SQL outputs (zero-shot) before and after supervised fine-tuning (SFT), using a small sample of target queries: two per database from the BIRD development set and 1% of the Gretel test set. Despite reduced sample size, relative alignment trends remain consistent with those from full datasets.

| Model | Base KL-Alignment | | Change After SFT on BIRD | | Change After SFT on Gretel | |
|---|---|---|---|---|---|---|
| | BIRD | Gretel | $\Delta$ BIRD | $\Delta$ Gretel | $\Delta$ BIRD | $\Delta$ Gretel |
| CodeLlama 13B | 0.38 | 0.49 | +0.08 | –0.10 | +0.00 | +0.05 |
| CodeLlama 7B | 0.36 | 0.50 | +0.07 | –0.16 | –0.01 | +0.05 |
| QWen 7B | 0.43 | 0.53 | +0.03 | –0.08 | +0.07 | +0.01 |
| QWen 1.5B | 0.45 | 0.54 | –0.04 | –0.08 | –0.07 | +0.01 |
| QWen 0.5B | 0.41 | 0.51 | +0.02 | –0.05 | –0.08 | +0.04 |

### 4.10 EXAMPLES OF QUERY CHANGES POST-SFT

To better understand the impact of SFT on model generation, we analyzed traceable features in queries generated by QWen-7B and CodeLlama-7B before and after SFT. We selected queries from the Gretel Test set where the base model generated correct outputs but the SFT model did not. After SFT on BIRD Train, systematic changes were observed in the models' use of aggregation functions, case expressions, and subqueries (see Appendix G). Notably, the frequency of pattern `att, SUM(exp)` (e.g., total balance per customer) significantly dropped while that of `SUM(exp)` (e.g., the total balance of all customers) significantly increased. Similarly, the frequency of `COUNT(*)` decreased and that of `COUNT(att)` increased. These changes closely mirrored the frequency of patterns in the training data. QWen-7B simulated these patterns more faithfully than CodeLlama-7B. Both models followed training data patterns after SFT, and when these patterns misaligned with the target, the likelihood of generating incorrect queries increased.

## 5 DISCUSSION

While proposing a novel dataset optimization method is not within the scope of this study, our findings offer valuable insights and guidelines for selecting and aligning datasets to improve NL2SQL model generalization:

**When Alignment is Meaningful**: As a syntactic metric, KL-alignment has its limitations. A model may generate sequences that align well with the target distribution, yet fail to produce valid SQL queries or accurately map natural language questions to SQL. In our evaluation, execution accuracy for base models demonstrated a strong correlation with KL-alignment (r = 0.941 for the QWen family and r = 0.921 for the CodeLlama family), but this correlation weakened for SFT models (r =

0.674 and r = 0.623, respectively). After filtering out configurations with $AR \leq 1$, the correlation significantly improved (r = 0.861 for QWen and r = 0.920 for CodeLlama), indicating that alignment is more meaningful and predictive of performance gains when $AR > 1$.

**Maximizing KL-Alignment**: When multiple SFT datasets are available, selecting the one with the highest KL-alignment to the target (ground truth) data is likely to improve model performance, as it ensures better alignment with the desired target data distribution.

**Fallback Strategies for Low Alignment:** When no dataset aligns well, practitioners can employ several strategies: (i) few-shot prompting using examples drawn from the most similar cluster to the target domain, (ii) targeted data synthesis to fill low-frequency structural patterns, and (iii) active curation or filtering of candidate datasets to reduce annotation costs. These strategies help maintain performance even when structural alignment is suboptimal.

**Cautiously Using Few-Shot Prompting**: Careful selection of few-shot examples can help maintain or enhance performance levels while minimizing the need for extensive labeled datasets. However, with smaller fine-tuned models such as Qwen 0.5B, few-shot prompting may exert a disproportionately high influence on output, potentially leading to unexpected outcomes, such as decreased KL-Alignment.

**Domain Generality:** While our study focuses on NL2SQL, the KL-alignment formulation only requires a compositional output space. It could be extended to code generation, data-to-text, or structured reasoning tasks, where program sketches or logical forms can serve as alignment templates. Future work could explore semantic alignment metrics to complement structural measures.

## 6 CONCLUSION

We investigated the problem of *dataset alignment* and its critical impact on the effectiveness of SFT for NL2SQL models. Through our KL-Alignment metric, we quantified how closely the structure of SFT training data matches that of target queries, and showed that high alignment leads to substantial gains in accuracy and generalization across domains. In contrast, poorly aligned datasets yield minimal improvements or even degrade performance, highlighting the importance of alignment-aware data selection in transfer learning pipelines.

Our study provides actionable insights: practitioners can pre-compute AR on candidate SFT datasets to guide selection, filter low-alignment subsets, and prioritize high-alignment data for efficient adaptation. When alignment is low, few-shot prompting, data synthesis, or active curation can mitigate performance loss.

Our study suggests that structural alignment between training and target distributions is a key lever for building robust, domain-adaptable NL2SQL systems. Future work could explore automated techniques for alignment-driven sampling or curriculum design to reduce manual overhead and improve cross-domain transferability. Additionally, while our experiments focus on small to mid-sized LLMs—commonly favored in resource-constrained settings—further investigation is needed to assess the extent to which these insights generalize to larger models.

Finally, although KL-Alignment captures distributional similarity over SQL syntax patterns, future extensions may incorporate semantic dimensions such as query correctness, schema grounding, and user intent, to provide a more comprehensive measure of dataset suitability for SFT.

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

## A  ABSTRACT SYNTAX TREE OF QUERIES

The abstract syntax tree (AST) of queries can be obtained using tools such as sqlglot [1]. For example, consider the following SQL query with its abstract syntax tree (AST) shown in Figure 4:

```
SELECT meal/enrollment FROM frpm WHERE county='Alameda' ORDER BY
(CAST(meal AS REAL) / enrollment) DESC LIMIT 1
```

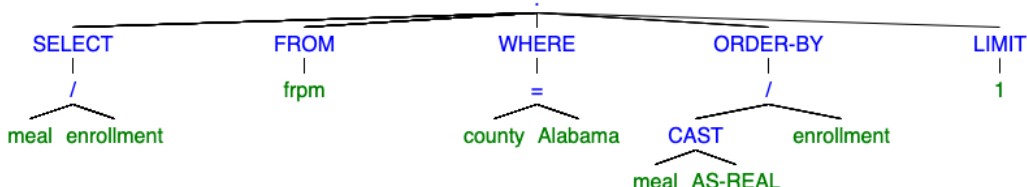

Figure 4: Abstract syntax tree (AST) of the given SQL query

By removing the leaf nodes from the AST, the query is transformed into the following structural template:

```
SELECT / FROM WHERE = ORDER BY ( CAST ( ) / ) DESC LIMIT.
```

## B  EVALUATION METRICS DETAILS

A standard metric for evaluating text-to-SQL models is execution accuracy (EX), as used in various benchmarks such as BIRD (Li et al., 2024) and Spider (Yu et al., 2018). Given a ground-truth SQL query and a predicted SQL query, the execution accuracy compares the execution results of the two queries on a database instance. If both queries produce identical results, the predicted query is considered correct, and the execution accuracy is marked as 1; otherwise, it is marked as 0. This metric is commonly used as it provides a comprehensive overview of model performance ignoring syntactic differences between queries.

Recognizing that two different queries may yield identical results on a database instance by chance, we additionally assess exact match accuracy (EM), which treats each clause as a set and compares the prediction for each clause to its corresponding clause in the ground truth query. A predicted SQL query is considered correct only if all of its components match the ground truth. This metric serves as a stricter variant of accuracy, as a predicted query can be correct but its clauses may not directly match those of the gold query, leading to a failure in exact match accuracy.

We assess dataset alignment mainly using KL-alignment, as introduced in §3.3. We also report the proportion of query templates in the target set that appear in the (training or model-generated) dataset, referred to as the OVLP ratio, as an alternative metric, provided in Appendix C.

## C  ALIGNMENT IN TERMS OF COMMON QUERY TEMPLATES

Table 5 presents the OVLP ratio, which quantifies the fraction of common query templates between datasets and model outputs. The results indicate that our tested models exhibit stronger alignment with Gretel compared to BIRD. Also, there is a notable alignment between the training and development/test sets within the same datasets.

Table 6 shows the impact of SFT on model alignment in terms of the OVLP ratio. Fine-tuning on BIRD substantially improves alignment with BIRD targets (e.g., CodeLlama 13B improves from

---

[1] https://github.com/tobymao/sqlglot

Table 5: Alignment of base models and train data sets in terms of OVLP ratio

|  | BIRD Dev | Gretel Test |
|---|---|---|
| CodeLlama 13B | 0.01 | 0.21 |
| CodeLlama 7B | 0.09 | 0.23 |
| QWen 7B | 0.06 | 0.22 |
| QWen 1.5B | 0.03 | 0.21 |
| QWen 0.5B | 0.03 | 0.17 |
| BIRD Train | 0.32 | 0.02 |
| Gretel Train | 0.02 | 0.61 |
| SmGretel Train | 0.00 | 0.17 |

0.01 to 0.21), but often reduces alignment with Gretel. In contrast, SFT on Gretel yields consistent gains in alignment with Gretel across all models, while maintaining or only slightly reducing alignment with BIRD.

Table 6: Change in structural alignment before and after supervised fine-tuning (SFT), measured via the OVLP ratio (the fraction of predicted queries matching the structure of target queries). Base scores are shown alongside alignment improvements ($\Delta$) after SFT on BIRD and Gretel datasets.

| Model | Base OVLP Ratio | | Change After SFT on BIRD | | Change After SFT on Gretel | |
|---|---|---|---|---|---|---|
| | BIRD | Gretel | $\Delta$ BIRD | $\Delta$ Gretel | $\Delta$ BIRD | $\Delta$ Gretel |
| CodeLlama 13B | 0.01 | 0.21 | +0.20 | −0.08 | +0.10 | +0.13 |
| CodeLlama 7B | 0.09 | 0.23 | +0.11 | −0.21 | −0.04 | +0.10 |
| QWen 7B | 0.06 | 0.22 | +0.11 | −0.15 | +0.08 | +0.03 |
| QWen 1.5B | 0.03 | 0.21 | +0.09 | −0.18 | −0.03 | +0.02 |
| QWen 0.5B | 0.03 | 0.17 | +0.11 | −0.10 | −0.03 | +0.03 |

These results confirm that models adapt structurally to the training domain, with improvements in OVLP ratio indicating better structural generalization to the target query distribution. Notably, large models like CodeLlama 13B benefit more symmetrically from domain-specific fine-tuning than smaller QWen variants, which show trade-offs between domains.

## D CHANGE IN ALIGNMENT AFTER SFT ON GRETEL

Table 7: KL-alignment before and after SFT on Gretel datasets. Left: Baseline KL-alignment scores (higher is better) with BIRD (dev) and Gretel (test). Center: Change ($\Delta$) in alignment after fine-tuning on the full Gretel training set. Right: Change after fine-tuning on the smaller SmGretel subset. Positive values indicate improved alignment with the respective dataset.

| Model | Base KL-Alignment | | Change After SFT on Gretel | | Change After SFT on SmGretel | |
|---|---|---|---|---|---|---|
| | BIRD | Gretel | BIRD | Gretel | BIRD | Gretel |
| CodeLlama 13B | 0.51 | 0.64 | +0.02 | +0.13 | +0.02 | +0.11 |
| CodeLlama 7B | 0.49 | 0.68 | -0.01 | +0.09 | -0.01 | +0.06 |
| QWen 7B | 0.61 | 0.68 | +0.06 | +0.02 | -0.02 | +0.06 |
| QWen 1.5B | 0.60 | 0.69 | -0.09 | +0.03 | -0.09 | +0.03 |
| QWen 0.5B | 0.57 | 0.66 | -0.13 | +0.00 | -0.11 | +0.03 |

Table 7 shows that SFT on Gretel improves KL-alignment with the Gretel test set across all models, confirming that fine-tuning effectively adapts model outputs to the target domain. Larger models like CodeLlama 13B gain the most (+0.13), while even smaller models show modest improvements. This pattern holds for both the full Gretel and smaller SmGretel training sets, suggesting that even limited in-domain data can drive meaningful structural alignment.

In contrast, alignment with the BIRD dataset often decreases after SFT, especially for smaller QWen models (e.g., –0.13 for QWen 0.5B), indicating a loss in generalization. Larger models like CodeLlama 13B maintain or slightly improve BIRD alignment, reflecting stronger generalization capacity. SmGretel tends to preserve BIRD alignment better than the full Gretel set, highlighting its potential for efficient fine-tuning with less risk of overfitting.

## E   FEW-SHOT EXAMPLES

Here are the few-shot examples used in the first round (ExS1) and the second round (ExS2):

```
-- ExS1
SELECT s.sname, a.album_name
FROM singer s JOIN album a ON s.singer_id = a.singer_id
WHERE s.nation = 'USA';

SELECT s.sname, s.age
FROM singer s JOIN album a ON s.singer_id = a.singer_id
WHERE a.genre = 'Rock';

SELECT AVG(s.salary)
FROM singer s JOIN album a ON s.singer_id = a.singer_id
WHERE s.nation = 'Japan' AND s.age BETWEEN 30 AND 40 AND a.release_year
    >= (s.year - 5);

-- ExS2
SELECT s.sname, a.album_name
FROM singer s JOIN album a ON s.singer_id = a.singer_id
WHERE s.nation = 'USA';

SELECT CAST(
COUNT(CASE WHEN T3.gender = 'M' THEN 1 ELSE NULL END) AS REAL) * 100 /
    COUNT(T2.person_id)
FROM noc_region AS T1 INNER JOIN person_region AS T2 ON T1.id =
    T2.region_id
INNER JOIN person AS T3 ON T2.person_id = T3.id
WHERE T1.region_name = 'Estonia';

SELECT name, growth_rate
FROM (SELECT name, growth_rate, ROW_NUMBER() OVER (ORDER BY growth_rate
    DESC) rn
FROM marine_species) t
WHERE rn <= 3;
```

## F   FINE-TUNING PROMPT

Here, we provide an example of a fine-tuning prompt used in the supervised fine-tuning process.

Listing 1: Example of fine-tuning prompt

```
CREATE TABLE region (
    region_id INT,
    region_name STRING,
    PRIMARY KEY (region_id)
);

CREATE TABLE timber (
    timber_id INT,
    region_id INT,
    year_time INT,
    volume INT,
    PRIMARY KEY (timber_id),
    FOREIGN KEY (region_id) REFERENCES region(region_id)
);

CREATE TABLE wildlife (
    wildlife_id INT,
    region_id INT,
    species_count INT,
    PRIMARY KEY (wildlife_id),
    FOREIGN KEY (region_id) REFERENCES region(region_id)
);

-- External Knowledge:
-- Using valid SQLite and understanding External Knowledge, answer
-- the following questions for the tables provided above:
-- What is the total volume of timber produced by each region, along with
-- the total number of wildlife species in those regions, grouped by
   year?
```

## G   EXAMPLES OF QUERY CHANGES POST-SFT

Table 8 presents the changes in the frequency of queries exhibiting traceable patterns after supervised fine-tuning (SFT) on the BIRD Train dataset. The analysis focuses on cases where the base models, QWen-7B and CodeLlama-7B, originally generated correct queries but introduced errors post-SFT. The table highlights patterns that either increased ($\uparrow$) or decreased ($\downarrow$) in frequency relative to the base models, providing insight into the specific failure modes introduced by fine-tuning.

Table 8: Changes in the frequency of queries with those traceable patterns after SFT on BIRD Train for cases where the base models (QWen-7B and CodeLlama-7B) initially generated correct queries, but errors emerged after SFT. Arrows indicate an increase ($\uparrow$) or decrease ($\downarrow$) in frequency compared to the base models.

| Pattern | Frequency | | | | | Example |
|---|---|---|---|---|---|---|
| | QWen-7B | | CodeLlama-7B | | BIRD | |
| | Base | SFT | Base | SFT | Train | |
| `, SUM(exp)` | 129 | 13 $\downarrow$ | 120 | 72 $\downarrow$ | 35 | `SELECT region, SUM(amount)` |
| | | | | | | `FROM investments GROUP BY region` |
| `SUM(exp)` | 63 | 209 $\uparrow$ | 55 | 95 $\uparrow$ | 1168 | `SELECT SUM(amount) FROM investments` |
| `COUNT(*)` | 109 | 49 $\downarrow$ | 141 | 0 $\downarrow$ | 371 | `SELECT COUNT(*) FROM transactions` |
| `COUNT(att)` | 75 | 119 $\uparrow$ | 61 | 242 $\uparrow$ | 2861 | `SELECT COUNT(id) FROM transactions` |
| `CASE WHEN` | 8 | 40 $\uparrow$ | 4 | 33 $\uparrow$ | 776 | `SELECT SUM(CASE WHEN age < 18 THEN 1 ELSE 0)` |
| `IIF` | 0 | 10 $\uparrow$ | 0 | 0 | 162 | `SELECT SUM(IIF age < 18, NULL, salary)` |
| `UNION` | 2 | 23 $\uparrow$ | 1 | 9 $\uparrow$ | 22 | `SELECT ... UNION SELECT ...` |
| Subqueries | 19 | 69 $\uparrow$ | 70 | 65 $\downarrow$ | 723 | `SELECT name FROM (SELECT ...` |

