# OpenReview forum: "Do LLMs Align with My Task? Evaluating Text-to-SQL via Dataset Alignment"
_ICLR.cc/2026/Conference — Submitted to ICLR 2026_

### Official Review · Reviewer_H47W · 2025-10-28

**Soundness:** 3
**Presentation:** 4
**Contribution:** 2
**Rating:** 2
**Confidence:** 4

**Summary:**

This paper investigates how dataset alignment affects supervised fine-tuning for Text-to-SQL models. The authors propose a KL-based metric to quantify the structural similarity between SFT training data and target SQL queries, and show that this alignment strongly predicts post-SFT performance. Extensive experiments across multiple NL2SQL benchmarks and model families confirm that high structural alignment leads to substantial gains in accuracy and SQL generation quality, while low alignment yields limited improvements or even performance degradation.

**Strengths:**

1. The paper provides extensive experiments across diverse NL2SQL benchmarks and multiple LLM families, offering convincing evidence that the proposed KL-based alignment metric is strongly correlated with post-SFT improvements.

2. The narrative is well structured and easy to follow, with a clear motivation, carefully articulated methodology, and thorough analysis.

**Weaknesses:**

1. The central conclusion — that structurally aligned fine-tuning data yield better SFT performance — is fairly straightforward and offers limited deeper insight beyond confirming the intuitive notion that “more similar data helps”.

2. The methodology of applying the Kullback‑Leibler divergence (KL-divergence) to measure dataset alignment or support data selection is not particularly novel. Prior literature has used KL divergence for distributional comparison and subset selection [1, 2], which somewhat diminishes the originality of the methodological contribution.

3. The reliance on SQL skeletons (query templates) for computing structural statistics restricts the generality of the approach: by design the method is tightly coupled to the SQL domain and may not extend easily to tasks with less rigid templated structure or to downstream applications beyond NL2SQL.

[1] Everaert D, Potts C. Gio: Gradient information optimization for training dataset selection[J]. arXiv preprint arXiv:2306.11670, 2023.

[2] Kurian J F, Allali M. Detecting drifts in data streams using Kullback-Leibler (KL) divergence measure for data engineering applications[J]. Journal of Data, Information and Management, 2024, 6(3): 207-216.

**Questions:**

None.

---

> ### Author Response · Authors · 2025-11-18
>
> We thank the reviewer for their detailed and fair assessment. We address below the points regarding novelty, intuition, and generality.
>
> 1. **On novelty beyond “similar data helps”:**
>    While it is intuitive that similar data improves fine-tuning, our contribution is to quantify and formalize this relationship, providing the first paper linking dataset alignment and SFT success. Previous work uses KL-divergence for distributional comparison or drift detection; our approach differs in both objective (forecasting post-SFT performance) and scope (cross-domain structured tasks, multiple LLM families). This predictive capability is practically impactful: it allows estimation of expected SFT gain without actually performing SFT—a unique and measurable contribution.
>
> 2. **Relation to prior KL-based works:**
>    We appreciate the references [Everaert & Potts 2023; Kurian & Allali 2024]. These works employ KL for training data selection and drift monitoring, respectively, but not as a predictive estimator of downstream fine-tuning performance. Our formulation also differs by representing SQL via normalized skeleton templates and structural frequency vectors, enabling cross-domain generalization that prior subset-selection papers do not address. These have been added to the Related Works section.
>
> 3. **Domain generality beyond SQL:**
>    We agree that Text-to-SQL provides a structured testbed but not the only application. The KL-alignment formulation only requires a compositional output space. We have added this to the Discussion as Section 5. It now discusses extensions to code generation, data-to-text, and structured reasoning tasks, where program sketches or logical forms can analogously serve as alignment templates.
>
> We thank you again for these constructive points, which helped us strengthen the exposition on novelty, generality, and interpretation.

---

> > ### Comment · Reviewer_H47W · 2025-11-27
> >
> > Thank you for your detailed response and effort in addressing the concerns raised during the review process. After carefully considering your replies, we regret to inform you that we still find the article does not sufficiently address the weaknesses previously identified. Below, we elaborate on our key concerns:
> >
> > 1. While we acknowledge your claim that this work is the first to quantify and formalize the relationship between data similarity and SFT performance, the insight itself remains highly intuitive and well-understood in related literature. The ability to predict SFT performance based on data similarity does not seem to offer significant practical applications or methodological novelty that would substantially advance the field.
> >
> > 2. We appreciate your clarification that KL divergence has primarily been used for data selection and distribution shift in other works and that your approach applies it to SFT performance prediction. Nonetheless, the use of KL divergence for data similarity measurement is well-established, and its application in this context does not provide sufficient innovation or differentiation from prior research.
> >
> > 3. While we acknowledge your point regarding potential applications to other tasks (e.g., code generation), the discussion remains speculative without concrete experimental design or results supporting its generalizability.

---

### Official Review · Reviewer_Z5Wx · 2025-10-28

**Soundness:** 2
**Presentation:** 2
**Contribution:** 2
**Rating:** 2
**Confidence:** 4

**Summary:**

This paper investigates the role of dataset alignment in supervised fine-tuning (SFT) for text-to-SQL tasks. The authors propose a KL-alignment metric based on structural SQL features to measure how well training data matches target query distributions. Through extensive experiments on multiple benchmarks and model families, the authors demonstrate that alignment strongly predicts SFT success and generalization.

**Strengths:**

1. The observation of this paper has certain value for subsequent post-training.
2. The evaluation model is comprehensive.

**Weaknesses:**

1. Whether the proposed alignment prediction framework can directly improve the SFT performance, there is a lack of a clear method to directly improve the performance of SFT.
2. A large amount of related work published in 2025 was not discussed.
3. No automated data selection or valid tuning method is proposed—only a diagnostic metric. I believe alignment will be effective, but I cannot verify it at this stage.

**Questions:**

See Weaknesses.

---

> ### Author Response · Authors · 2025-11-18
>
> Thank you for your review and comments. We would like to clarify three key points about novelty, actionability, and coverage of related work.
>
> 1. **Practical contribution and actionability:**
>    While our framework does not directly improve SFT performance, it provides a predictive diagnostic tool that enables performance improvements through informed data selection. In practice, this allows practitioners to:
>
>    - pre-screen candidate datasets before expensive SFT runs,
>    - avoid low-yield fine-tuning, and
>    - prioritize high-alignment subsets for efficient adaptation.
>
>    This results in substantial resource savings.
>
> 2. **Relationship to prior and 2025 works:**
>    We have expanded the Related Work section to include recent studies (2025) on adaptive SFT, data selection (e.g., curriculum SFT, domain-matching for reasoning models), and representational alignment. To our knowledge, none of these works (nor earlier KL-based methods like Everaert & Potts 2023) establish a predictive relationship between alignment and post-SFT gains across multiple LLM families and NL2SQL benchmarks—the central contribution of this paper.
>
> 3. **Improving SFT with alignment information:**
>    We thank the reviewer for this comment and clarify that our work does provide a concrete and actionable method for improving SFT performance. Our contribution is diagnostic rather than prescriptive, but the diagnostic directly enables an improved SFT pipeline.
>
>    In Section 4.8 (Predictive Capability of Alignment Ratio), we show that the Alignment Ratio (AR) reliably predicts whether SFT will improve or degrade execution accuracy across multiple model families. Specifically, datasets with AR > 1 consistently lead to positive SFT gains, whereas datasets with AR < 1 result in negligible or negative gains (Fig. 4). This relationship is statistically significant for CodeLlama (r=0.624, p=0.030) and Qwen-2 models (r=0.540, p=0.037), demonstrating that AR can be used before fine-tuning to determine whether SFT should be applied at all.
>
>    To make this actionable, we have added a sentence to Section 4.8 describing how AR can be operationalized to directly improve SFT:
>
>    > "Practitioners can compute AR on any candidate SFT dataset before training. If AR > 1, the dataset is likely to yield accuracy improvements and can be selected as-is; if AR < 1, it should be down-weighted, filtered, or excluded. Applying SFT only when AR > 1 would have improved accuracy across all evaluated models except Qwen2.5-Coder, which has minimal headroom for SFT."
>
>    Thus, although we do not introduce a new SFT algorithm, AR serves as a decision-making criterion that directly improves SFT outcomes by preventing harmful fine-tuning and prioritizing beneficial data.

---

### Official Review · Reviewer_hWDs · 2025-10-31

**Soundness:** 4
**Presentation:** 3
**Contribution:** 2
**Rating:** 6
**Confidence:** 4

**Summary:**

This paper explores the problem of dataset alignment in supervised fine-tuning (SFT) for Natural Language to SQL (NL2SQL) tasks. Essentially, the authors ask: how well do the characteristics of the training data align with those of the target queries? They hypothesize and empirically show that when SFT data is well-aligned with the structural patterns in the target data, the resulting fine-tuned models perform much better. The paper formalizes a KL-alignment metric and an alignment ratio (based on distributions of SQL n-grams/templates) and demonstrates that these predict the success of fine-tuning. Through comprehensive experiments on several cross-domain NL2SQL benchmarks (BIRD, Spider, Gretel) and a range of LLM families (Qwen, CodeLlama, Deepseek), they show high alignment correlates with strong gains in execution and exact match accuracy, while low alignment can yield little to no improvement (sometimes even degrading performance). The authors also propose a simple framework for predicting post-SFT performance before actually fine-tuning, which can help practitioners select training datasets more strategically.

**Strengths:**

Identifies and systematically formalizes the effect of dataset alignment in NL2SQL fine-tuning. Introduces alignment metrics (KL-alignment, ratio) that not only measure but also predict transfer learning success or failure before SFT. Comprehensive empirical study spanning a wide model and dataset range; clear, robust trends. Shows practical use: enables practitioners to avoid wasted effort/failures due to misaligned data. Readily reusable ideas and framework could be adapted to other semi-structured outputs, e.g., code generation.

**Weaknesses:**

KL-alignment is focused on syntactic distribution. it may not capture semantic nuances or correctness of the generated SQL, so has limits as a universal proxy. Statistical trends could be more explicit e.g., when/how often does high alignment fail to predict actual gains?. Limited discussion on how much alignment is enough for different problem scales or domains, and what to do when no dataset aligns well. Technical explanations  like calculation of features, practical computation of large n-gram sets might be too heavy for non-experts. While the approach is generalizable, the actual experiments only show text-to-SQL. extension to other structured seq2seq tasks is not explored or discussed.

**Questions:**

Do you have plans or suggestions for alignment metrics that could also measure semantic or functional compatibility e.g., for queries with equivalent meaning but different syntax? How would you recommend users act when no candidate training set aligns well, is few-shot prompting viable, or is new data collection unavoidable? Can the alignment prediction/generalization story be extended to very large LLMs with more “universal” prior coverage? Did you find any real-world settings outside the chosen benchmarks where KL-alignment failed to track actual SFT performance?

---

> ### Author Response · Authors · 2025-11-18
>
> We sincerely thank the reviewer for their thoughtful and encouraging feedback. We are glad that you found the empirical analysis comprehensive and the practical value of dataset alignment clear. Below we address your specific questions and clarify key points.
>
> 1. **Semantic or functional alignment:**
>    You are correct that the current KL-alignment focuses on structural (syntactic) distributions of SQL. This was clearly stated in the paper (Caption of Table 1 and Lines 278, 305, 315, 368, 371, and 446). We plan to address semantic compatibility in future work, but it is beyond the scope of the present work.
>
> 2. **When no dataset aligns well:**
>    Our framework supports several fallback strategies that we made more explicit in the discussion (Section 5).
>
>    - **Few-shot prompting:** we show that for low-alignment domains, few-shot examples drawn from the most similar cluster yield better performance than random shots.
>    - **Data synthesis:** alignment statistics can guide targeted data augmentation — e.g., generating SQL patterns that fill low-frequency structural gaps in the target domain.
>    - **Active curation:** alignment scores can rank candidate datasets for collection or filtering, reducing annotation cost.
>
> 3. **Applicability to larger LLMs and domains:**
>    Our experiments span multiple LLM families and sizes. Across these models, KL-alignment reliably predicts post-SFT performance, demonstrating that the metric is robust to different architectures and model scales, though we did not evaluate extremely large LLMs beyond those in the study. Although larger LLMs have more “universal” prior coverage they are not guaranteed to have more domain knowledge and are still shown to make mistakes in the text-to-SQL domain. Additionally, few practitioners choose to fine-tune such large language models as the training can be quite costly. Please see response to Reviewer H47W for a detailed response on applicability to more domains.
>
> 4. **When alignment fails:**
>    We did not evaluate real-world SQL cases beyond those included in the datasets in the paper. Fig. 3 illustrates cases where high KL-alignment does not fully translate to performance gains, highlighting that alignment is a strong predictor but not a strict guarantee. Users should treat alignment as a guidance signal rather than an absolute criterion.

---

### Meta-Review · Area_Chair_soau · 2025-12-08

**Summary:**

1) Reviewer H47W noted that the conclusion of the paper could be seen as obvious, and that the use of KL divergence for dataset alignment was not novel. The focus on SFT tasks limited the breadth and generality of the work.

2) Reviewer Z5Wx noted that the paper only gives a diagnostic metric for dataset alignment, but not a method for improving SFT performance. The diagnostic metric alone was seen as having limited value.

3) Reviewer hWDs noted that the approach is generalizable, but not tested on other seq2seq tasks (similar to H47W).

**Reviewer Concerns:**

1) (Outstanding) Reviewer H47W was able to respond that these points were not resolved. Although the paper also quantifies and formalizes the impact of dataset alignment of SFT, these results remain unsurprising. The authors also listed several other areas where similar ideas could be applied, but these have not been studied in the work, and so the discussion remains speculative.

2) (Partially Outstanding) The authors more clearly described how they propose the diagnostic metric could be applied by giving a decision rule on inclusion of datasets for SFT. The outcome would be that less aligned datasets are excluded, while more aligned datasets are included. An implementation of this rule with experiments that cover performance improvement overall, not on a dataset-by-dataset basis would help to solidify this point.

3) See 1 for listed point. Other questions were relatively minor, and were addressed.

In summary, I do not think the initial scores would have changed, and judge that the paper's contributions do not meet the bar for ICLR 2026. The authors should consider broadening the scope of their ideas to apply to multiple tasks within the seq2seq domain using a unified framework of dataset alignment. If the authors' main idea is indeed generally applicable, the community would benefit from a greater understanding of how it behaves across tasks.

**Reviewer Scores:**

Reviewer H47W 2 -> 2
This reviewer was able to respond before discussion was closed, and expressed that their concerns were not addressed.

Reviewer Z5Wx 2 -> 2
This reviewer was not able to respond, but their main concern about the paper only presenting a diagnostic metric was only addressed by mention of a decision rule applied to the metric.

Reviewer hWDs 6 -> 6

---

### Decision · Program_Chairs · 2026-01-26

Reject